# Prevalence and determinants of neonatal near miss in Ethiopia: A systematic review and meta-analysis

**Ababe Tamirat Deressa**[1]*, **Melese Siyoum Desta**[2]

**1** School of Nursing, Saint Paul's Hospital Millennium Medical College, Addis Ababa, Ethiopia, **2** Department of Midwifery, College of Medicine and Health Sciences, Hawassa University, Hawassa, Ethiopia

* waktasuabe@gmail.com

**Data Availability Statement:** All relevant data are within the paper and its Supporting Information files.

## Abstract

### Introduction

Neonatal near miss is a condition of newborn infant characterized by severe morbidity (near miss), but survived these conditions within the first 27 days of life. It is considered as the first step to design management strategies that can contribute in reducing long term complication and mortality. The aim of this study was to assess prevalence and determinants of neonatal near miss in Ethiopia.

### Methods

The protocol of this systematic review and meta-analysis was registered at the Prospero with a registration number of (PROSPERO 2020: CRD42020206235). International online databases such as PubMed, CINAHL, Google scholar, Global Health, Directory of open Access journal and African Index Medicus were used to search articles. Data extraction was undertaken with Microsoft Excel and STATA11 was used to conduct the Meta-Analysis. Random effect model analysis was considered when there was evidence of heterogeneity between the studies.

### Results

The overall pooled prevalence of neonatal near miss was 35.51% (95%CI: 20.32–50.70, $I^2$ = 97.0%, p = 0.000). Primiparity (OR = 2.52, 95%CI: 1.62, 3.42), referral linkage (OR = 3.92, 95%CI: 2.73, 5.12), premature rupture of membrane (OR = 5.05, 95%CI: 2.03, 8.08), Obstructed labor (OR = 4.27, 95%CI: 1.62, 6.91) and maternal medical complications during pregnancy (OR = 7.10, 95%CI: 1.23, 12.98) had shown significant statistical association with neonatal near miss.

### Conclusion

The prevalence of neonatal near miss in Ethiopia is evidenced to be high. Primiparity, referral linkage, premature rupture of membrane, obstructed labor and maternal medical complications during pregnancy were found to be determinant factors of neonatal near miss.

**Funding:** The authors received no specific funding for this work

**Competing interests:** The authors have declared that no competing interests exist.

## Introduction

Neonatal near miss is defined as a condition of newborn infant characterized by severe morbidity (near miss) of pragmatic and management criteria, but survived these conditions within the first 27 days of life. The pragmatic criteria includes birth weight < 1750g, Apgar score <7 at 5 minute and gestational age <33 complete weeks. The management criteria involves parenteral antibiotic therapy, nasal CPAP, any intubation, phototherapy within 24 hours of life, cardiopulmonary resuscitation, use of vasoactive drugs, use of anticonvulsants, use of surfactant, use of blood products, surgery and use of steroids for the treatment of refractory hypoglycemia [1, 2]. For neonates with life threatening situations, neonatal near miss was proposed as a tool for assessing the quality of care. It is also considered as the first step to design management strategies that can contribute in reducing long term complication and mortality [3].

Wide range rates of neonatal miss was reported across the world. Almost finding of different studies agree that the rate and/or prevalence and/or incidence of neonatal near miss is higher than that of infant or neonatal mortality. For instance, WHO multicounty Survey on Maternal and Newborn Health reported 72.5 and 9.2 neonatal near miss and early neonatal death per 1000 livebirths respectively [4]. In Brazil survey, the rate of neonatal near miss and early neonatal mortality per 1000 live births were 39.2 vs 11.1 [5].

Studies in Brazil reported different prevalence or rates of neonatal near miss from different parts of the country. The stated findings were 21.4, 91.7, 39.2 and 33 neonatal near miss cases per 1,000 live births in Brazil. In all of these studies neonatal mortality was less prevalent than neonatal near miss [5–8]. The study in Nepal indicated the prevalence to be 7.9% [9]. The Fifth Perinatal Care Survey of South Africa reported the prevalence of neonatal near miss and neonatal mortality to be 2.47 and 0.63 respectively [10]. In tertiary hospitals of Southern Ghana the prevalence of neonatal near miss was 69.5% [11] and 12% incident in Obafemi Awolowo University teaching hospitals of Nigeria [12].

Various factors contribute to the development of neonatal miss. For example, delays in obstetric care, inadequate prenatal care, delayed access to health services [13]; advanced maternal age, multiparty, hypertensive disease, forceps-assisted vaginal delivery [7]; gestational age of less than 32 weeks, use of mechanical ventilation, congenital malformation [5]; gestational age < 33 weeks, neurologic dysfunction, respiratory dysfunction, hemoglobin < 10 g/dl [11], not being delivered with cesarean section, and severe maternal morbidity [9] are some of the stated factors from different studies.

In Ethiopia, the neonatal and infant mortality rate remain higher than the rate from some developing countries. The recent finding reported the neonatal mortality rate of 29.1 [14] and the infant mortality rate of 48 deaths [15] per 1,000 live births respectively.

As it has been evidenced in different findings, it can be understood that we can save the life many neonates from those who were diagnosed for neonatal near miss. Hence, Identifying the burden of neonatal near miss and its associated factors directly contribute to reduction of neonatal mortality since it help us to early diagnosis and treat the causes the mortality. However, different studies reported different prevalence and determinant factors of neonatal near miss in the country. Hence, it remained unclear to understand the burden and determinants of neonatal near miss at country level. Thus, this study aimed at generating single study that estimates the prevalence and determinants of neonatal near miss in Ethiopia.

## Methods

### PROSPERO registration

The protocol of this systematic review and meta-analysis was registered at the Prospero with a registration number of (PROSPERO 2020: CRD42020206235) that is available from https://www.crd.york.ac.uk/PROSPERO.

### Search strategy

International online databases (such as PubMed, CINAHL, Google scholar, Global Health, Directory of open Access journal and African Index Medicus) and the Hawassa university library were used to search articles on prevalence and determinants of neonatal near miss from August 15–25, 2020. To access all important articles from the mentioned data bases, searching terms were based on adapted PICO questions. Different approaches were utilized to get comprehensive data on neonatal near miss. MeSH terms including "neonatal near miss" OR "perinatal asphyxia OR "birth asphyxia" OR "prematurity" OR "preterm birth" OR "neonatal jaundice" OR "neonatal hypoglycemia" OR "Neonatal organ failure" OR "Neonatal seizure" OR "severe congenital anomaly" AND related in Ethiopia were searched in the international online electronic database. These MeSH terms are generated based on the definition on neonatal near miss as stated in the introduction.

### Inclusion and exclusion criteria

Cross-sectional, case-control and cohort studies were included (Table 1). Cross sectional studies were only considered to calculate the pooled prevalence of neonatal near miss. Research articles written in English language which reported the prevalence or magnitude and/or associated factors or predictors or determinants of neonatal near miss were included in this Systematic review and meta-analysis. On the other hand, articles without full text and abstract, duplicated studies and anonymous reports were excluded.

### Data extraction and quality assessment

After collecting findings from all databases and screening the eligible studies, the articles were extracted on Microsoft Excel spreadsheet. The two authors (ATD & MSD) independently extracted the data and reviewed all the screened and included articles. Any disagreement was handled by the third invited reviewer. After all, a consensus was reached through discussion between authors. Newcastle- Ottawa Quality Assessment Scale (NOS) for cross- sectional, case-control and cohort studies was used to assess the methodological quality of each study and to deter- mine the extent of addressing bias in its selection, comparability and outcome measurement. As per the NOS, fulfilling 3 or 4 criteria in selection domain and 1 or 2 in

**Table 1. Study characteristics included in the systematic review and meta-analysis.**

| S/No | Author [year] | Region | Sample Size | Study design | Data collection techniques | Study Setting |
|------|---------------|--------|-------------|--------------|----------------------------|---------------|
| 1. | Belay HG et al [from Research square] | Amhara | 404 | Cross sectional | Checklist based | Institution based (hospital) |
| 2. | Mersha A et al [2019] | SNNPR* | 484 | Case-Control | Interviewer administered | Institution based (hospital) |
| 3. | Tekelab T et al [2020] | SNNPR* | 2,704 | Cohort study | Interviewer administered | Institution based (hospital) |
| 4. | Woldeyes Y et al [unpublished] | SNNPR* | 380 | Cross sectional | Checklist based | Institution based (hospital) |
| 5. | Tassew HA et al [2020] | Amhara | 422 | Cross sectional | Checklist based | Institution based (hospital) |

*Southern Nation, Nationalities and People's Region

comparability domain and 2 or 3 in outcome/exposure domain guarantees the study to be categorized as good quality study. Accordingly, all included studies are with good quality.

### Measurement of outcome

Two main outcome variables were considered in this study. The first is prevalence of neonatal near miss and the second is determinants of neonatal near miss. Standard error (SE) and Odds ratio (OR) were calculated as effect measure for the analysis of prevalence and determinant factors respectively.

### Publication bias and heterogeneity

To check the heterogeneity of the studies, the analysis of Cochrane Q test and $I^2$ with its corresponding p-value were used. A value of 0, 25, 50, and 75% was used to declare the het- erogeneity test as no, low, medium and high heterogeneity respectively. Analysis of random effect model computed when there is evidence of heterogeneity. Funnel plot and Egger regression asymmetry tests were employed to assess the existence of publication bias. A funnel plot showed that asymmetrical distribution and the Egger test value was 0.034. Hence, there is publication bias in the studies.

### Data analysis

The data were extracted using Microsoft Excel. Then, meta-analysis were conducted using STATA 11 software. The estimated prevalence of each study was presented using forest plots with standard error as effect measure and 95% confidence interval (CI). Besides, the analysis of determinant factors from the studies were presented by using forest plots with effect measure of Odds ratio (OR) and its 95% confidence interval (CI).

### Ethical consideration

Ethical clearance is not needed for this Systematic Review and Meta-Analysis.

## Results

A total of 101 articles and one article were found with electronic and other approach of searching respectively. Twenty eight were duplication and 74 articles were screened. From these 74 articles, 69 of them were excluded due to irrelevance, absence of full text or Abstract. Finally, 5 articles were remained eligible for this systematic review and meta-analysis among which three articles were used to determine the pooled prevalence of neonatal near miss (**Fig 1**).

### Epidemiology of neonatal near miss in Ethiopia

A wide-ranging prevalence of neonatal near miss was observed across different studies included in this review [16–18]. The overall pooled prevalence of neonatal near miss in Ethiopia was found to be 35.51% (95%CI: 20.32–50.70, $I^2$ = 97.0%, p = 0.000) using a random effect model (**Fig 2**).

### Determinants of neonatal near miss

Five studies were used to analyze determinant factors for neonatal near miss (**Fig 3**).

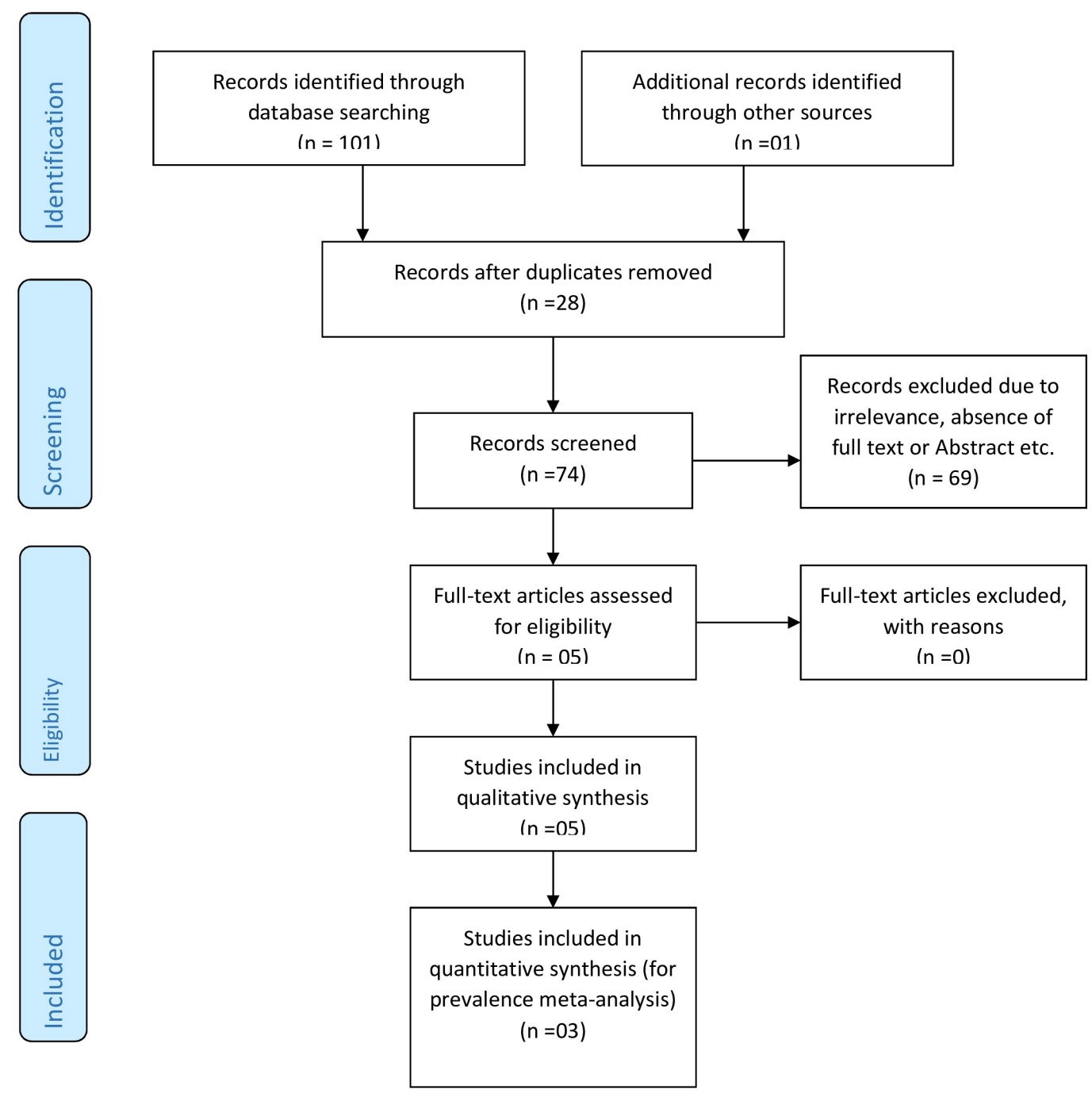

**Fig 1. PRISMA flow chart of study selection for systematic review and meta-analysis of prevalence of neonatal near miss in Ethiopia.**

### Association between primiparity and neonatal near miss

Two studies were involved for meta-analysis of this category [16, 17]. The likelihood of developing neonatal near miss among neonates who were born to primipara mother was 2.52 times (OR = 2.52, 95%CI: 1.62, 3.42) more likely than their counter parts. Studies included in this meta-analysis were characterized by no heterogeneity ($I^2 = 0.0\%$, p = 0.829). Thus, the analysis was undertaken with fixed effect model (**Fig 4**).

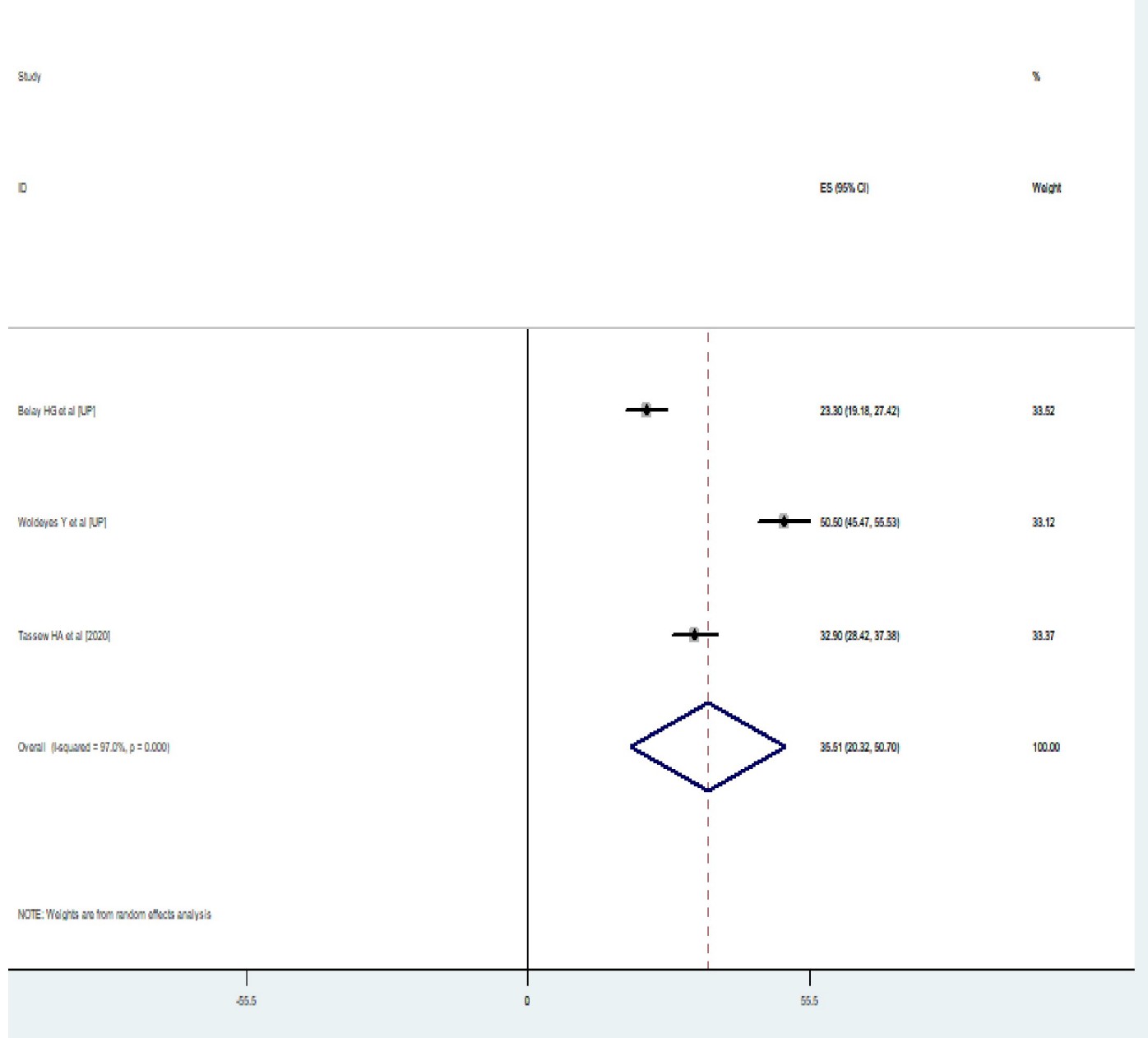

**Fig 2. Pooled prevalence of neonatal near miss in Ethiopia.**

### Association between referral linkage and neonatal near miss

Two studies were involved in this category of meta-analysis [16, 19]. The likelihood of developing neonatal near miss among neonates who had history of referral linkage was almost 4 folds (OR = 3.92, 95%CI: 2.73, 5.12) higher than their counter parts. Studies included in this meta-analysis were characterized by no heterogeneity ($I^2$ = 0.0%, p = 0.502). Thus, the analysis was conducted with fixed effect model (**Fig 5**).

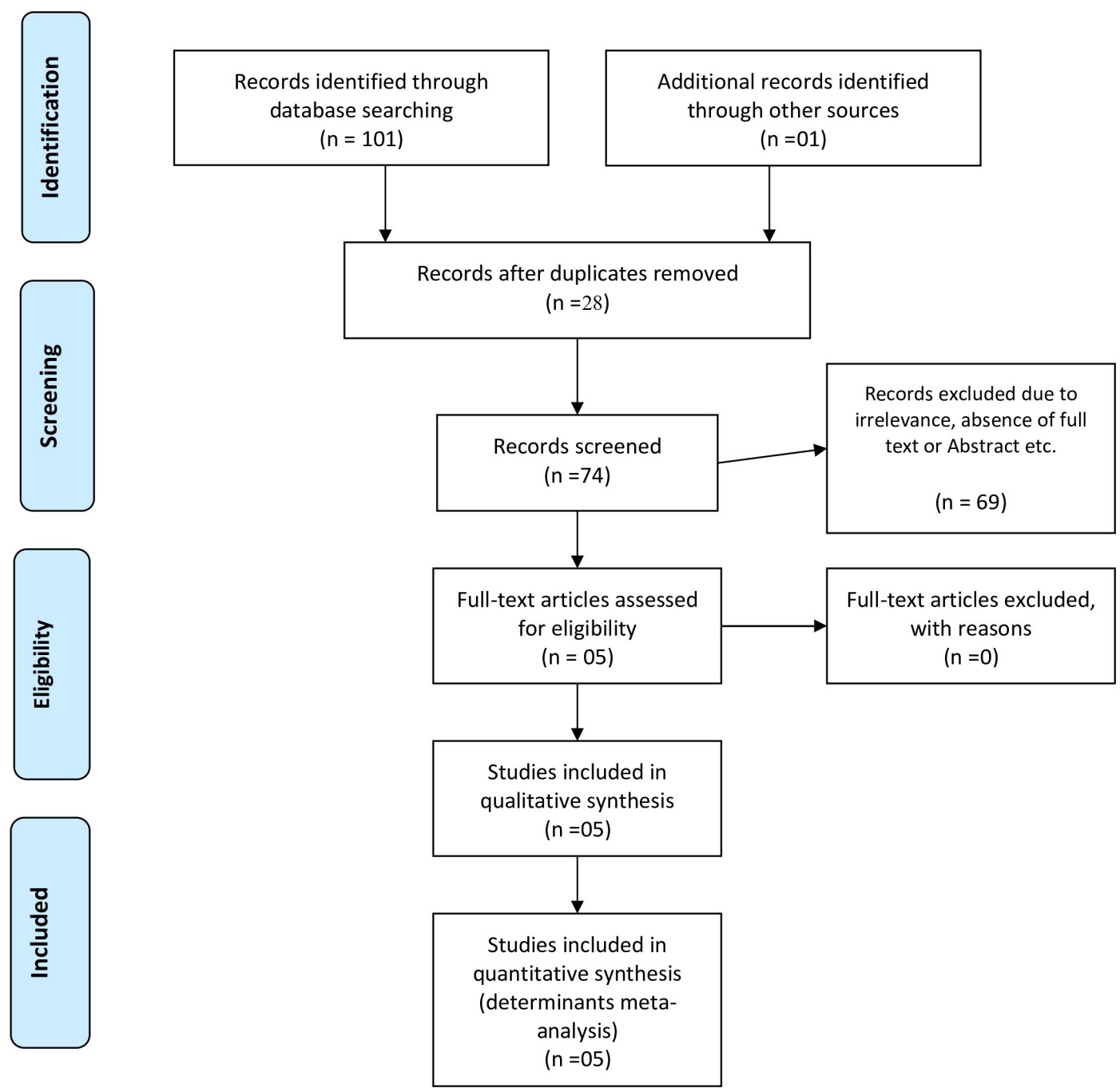

**Fig 3. PRISMA flow chart of study selection for systematic review and meta-analysis of determinants of neonatal near miss in Ethiopia.**

## Association between premature rupture of membrane (PROM) and neonatal near miss

Three studies were involved for meta-analysis of this category [16, 17, 20]. Neonates who were born after premature rupture of membrane (PROM) had 5 times (OR = 5.05, 95%CI: 2.03, 8.08) more odds of developing neonatal near miss than their counter parts. Studies included in this meta-analysis were characterized by moderate heterogeneity ($I^2$ = 49.1%, p = 0.140). Hence, the analysis was undertaken with random effect model (**Fig 6**).

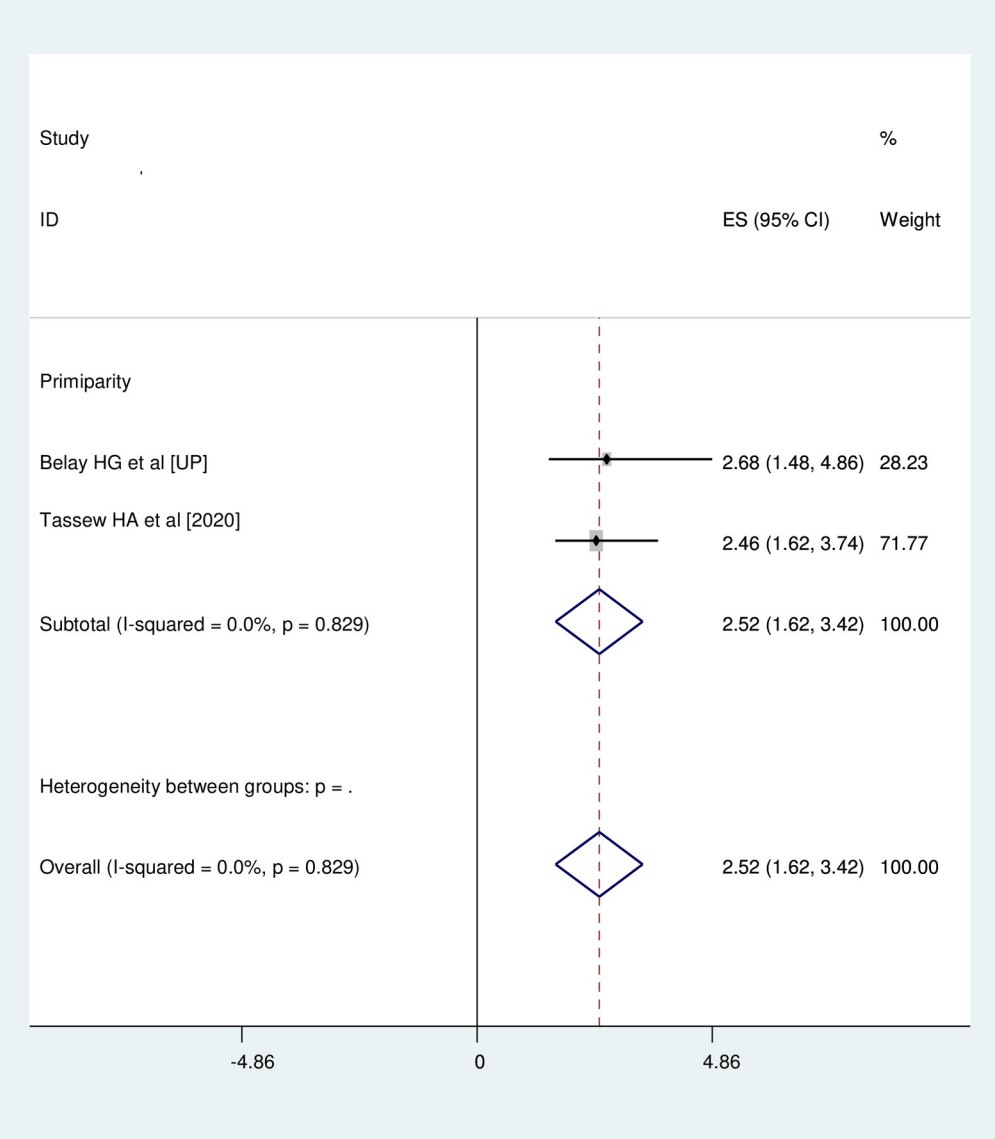

**Fig 4. Association between primiparity and neonatal near miss.**

## Association between obstructed labor and neonatal near miss

Two studies were involved in this category of meta-analysis [16, 17]. Neonates who were born to mothers with history of obstructed labor had 4.27 times (OR = 4.27, 95%CI: 1.62, 6.91) more odds of developing neonatal near miss than their counter parts. Studies included in this meta-analysis were characterized by no heterogeneity ($I^2$ = 0.0%, p = 0.908). Thus, analysis was undertaken with fixed effect model (**Fig 7**).

## Association between history of maternal medical complications during pregnancy and neonatal near miss

Three studies were involved for meta-analysis of this category [17, 19, 20]. The likelihood of developing neonatal near miss among neonates who were born to mothers who had history of

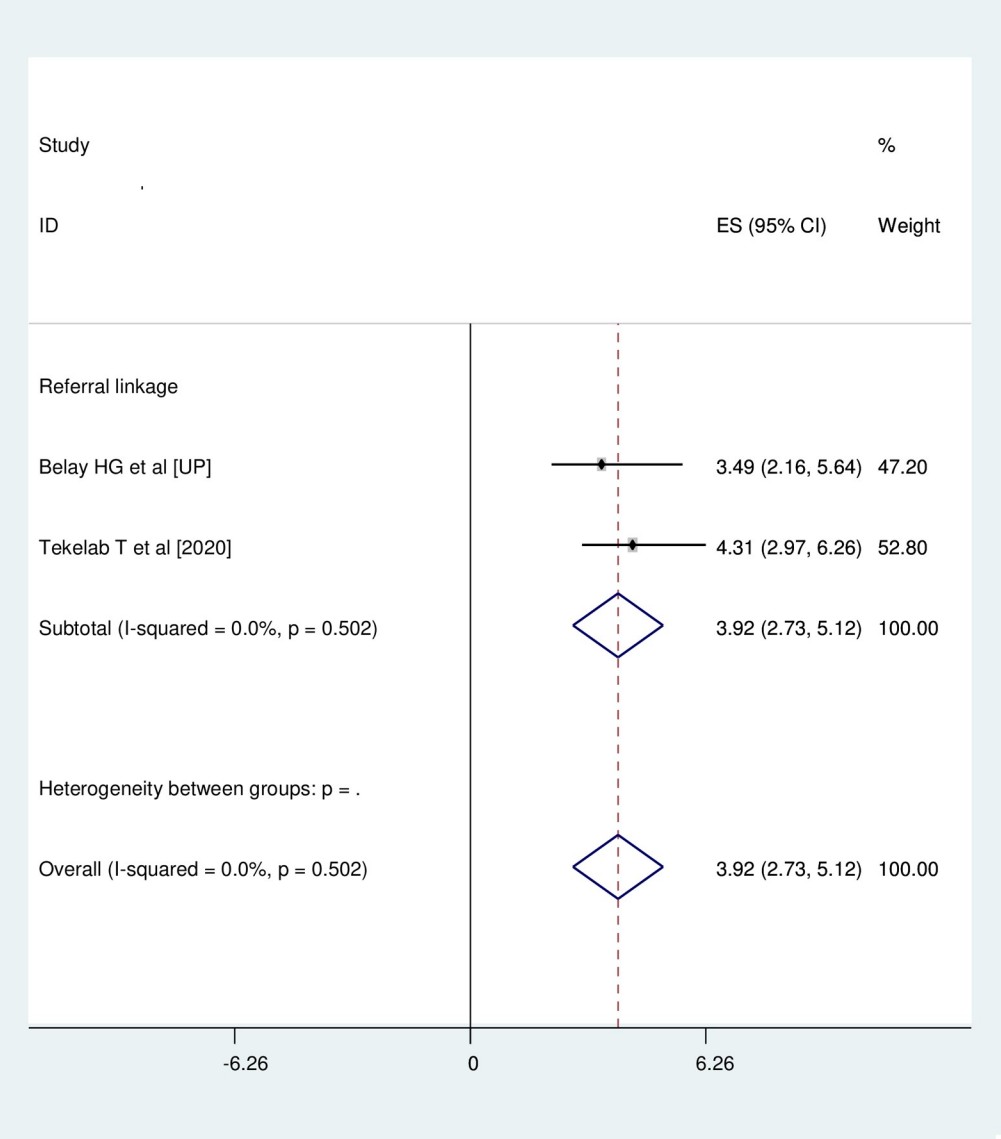

**Fig 5. Association between referral linkage and neonatal near miss.**

medical complications during pregnancy was 7 times (OR = 7.10, 95%CI: 1.23, 12.98) more likely than those neonates born to mothers who had no history of medical complications. Studies included in this meta-analysis were characterized by high heterogeneity ($I^2$ = 82.7%, p = 0.001). Thus, the analysis was undertaken with random effect model (**Fig 8**).

## Discussion

In this Systematic review and Meta-analysis, the prevalence and determinants of neonatal near miss in Ethiopia were explored. Accordingly, the finding evidenced as neonatal near miss is significant public health problem in the country so that policy maker should design strategies to improve obstetric and neonatal care for its reduction. The pooled prevalence of neonatal near miss in this study is higher than that of reported prevalence from Brazil [5–8], Nepal [9], south Africa [10], Ghana [11] and Nigeria [12]. These stated countries might

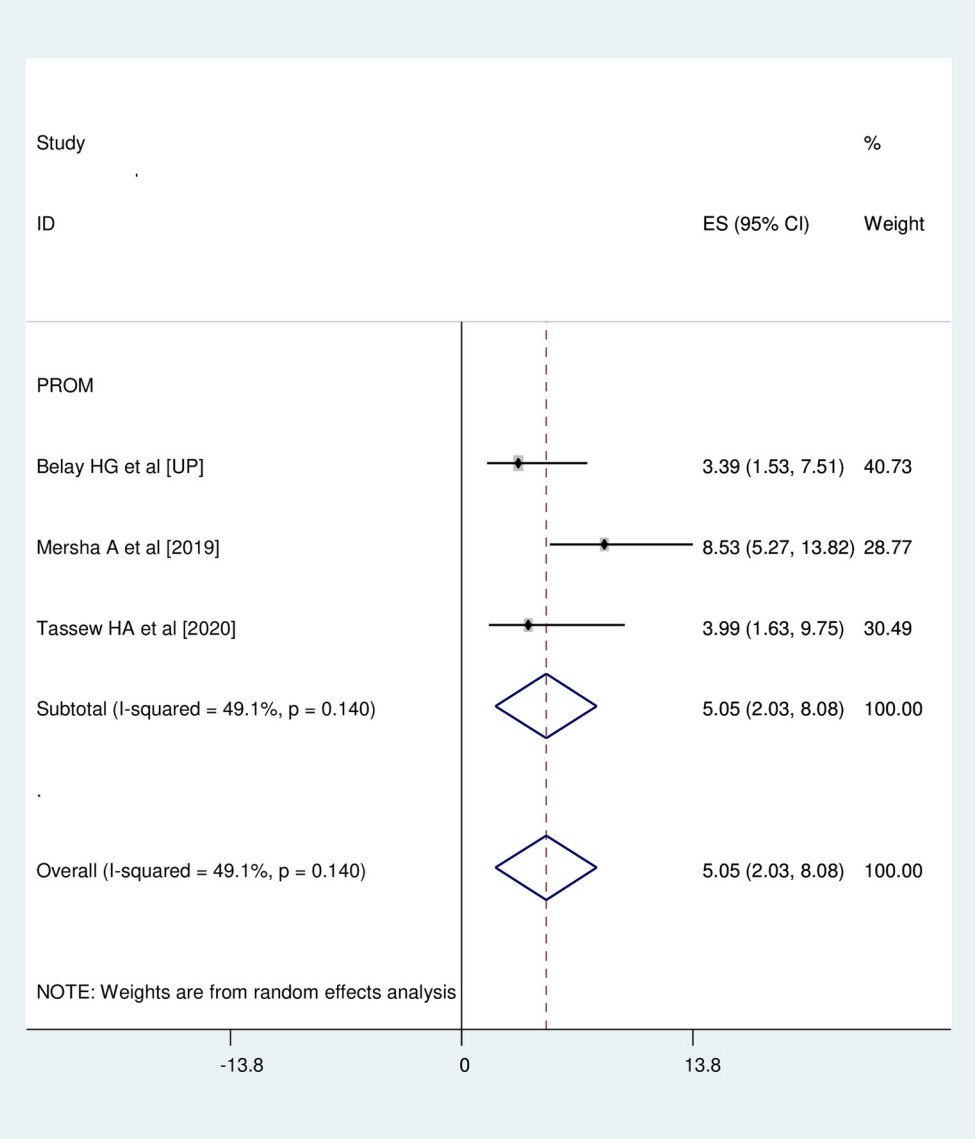

**Fig 6. Association between premature rupture of membrane (PROM) and neonatal near miss.**

have better socio-economic status that may contribute for having better facility that gives quality obstetric and neonatal care. This care is directly linked with the status of neonatal near miss in a country. Thus, the variation of the prevalence can be due to this differences. Moreover, other reasons for this variation may include differences in study settings, study design and sample size.

Primiparity was found to be risk for neonatal near miss in this study and neonates of primipara mothers shall get close attention in neonatal care. This finding is supportable with finding from Nepal where neonates who born to multiparous women less likely develop neonatal near miss in Nepal [9]. Primipara mothers may not have adequate birth preparedness as multipara mothers which highly contributes to prevent maternal and neonatal complications during and after pregnancy. Hence, this may be the reason behind for association of primiparity and neonatal near miss.

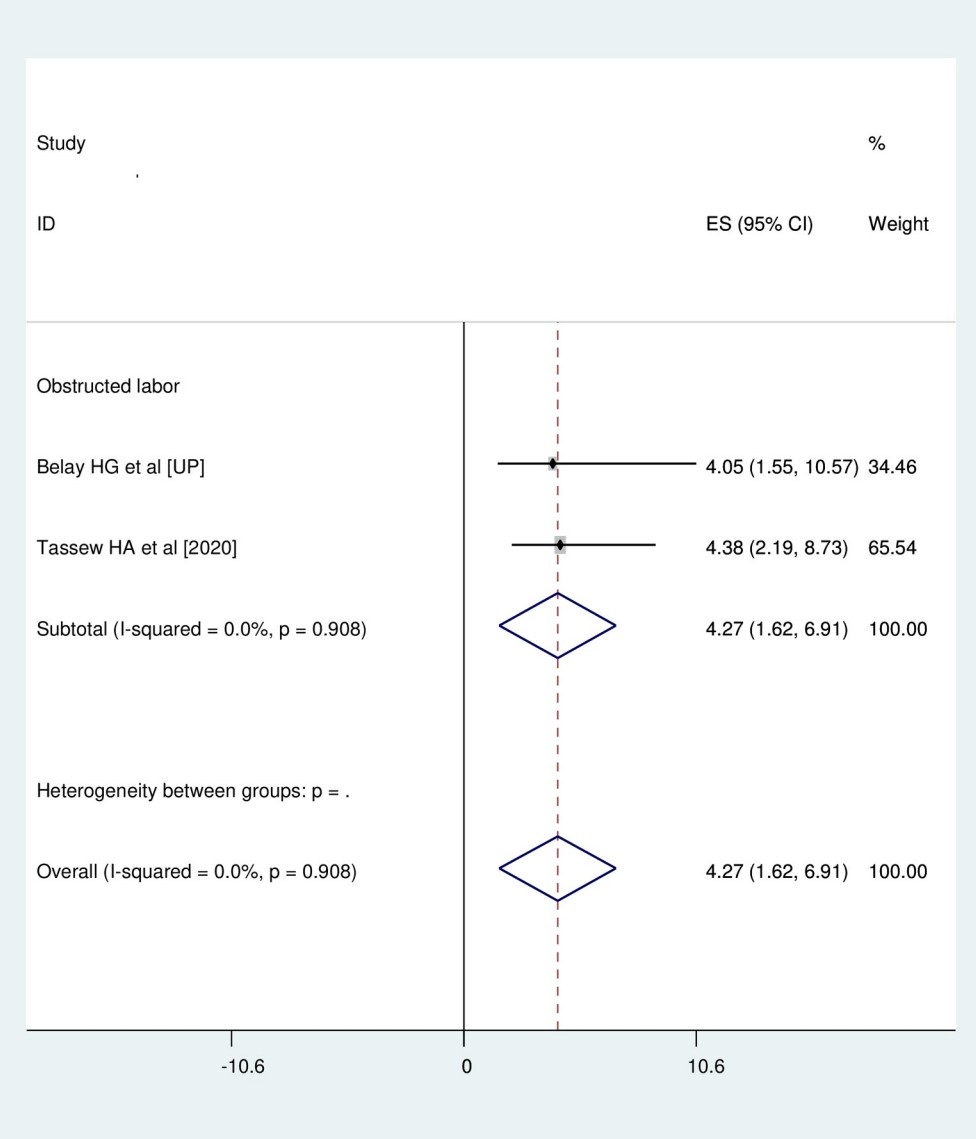

**Fig 7. Association between obstructed labor and neonatal near miss.**

In this study, neonates who had history of referral linkage showed more odds of developing neonatal near miss. This is may be due to delay of getting quality services that is evidenced to be the risk for developing neonatal near miss [12, 13]. In addition, the case/morbidity of the neonates may get aggravated till they reach the institutions to which they are referred. On the other hand, this study also evidenced as premature rupture of membrane (PROM) is one of the determinant factors for neonatal near miss. Different studies [21–24] stated as PROM is risk factor for neonatal sepsis. Neonatal sepsis is treated with parenteral antibiotic therapy which is one of the management criteria [1] to define neonatal near miss. Thus, this reason may justify the association between PROM and neonatal near miss. This association may imply as Ministry of health and Regional health bureau shall integrate Neonatal intensive care unit in each hospitals so as to reduce referral linkage.

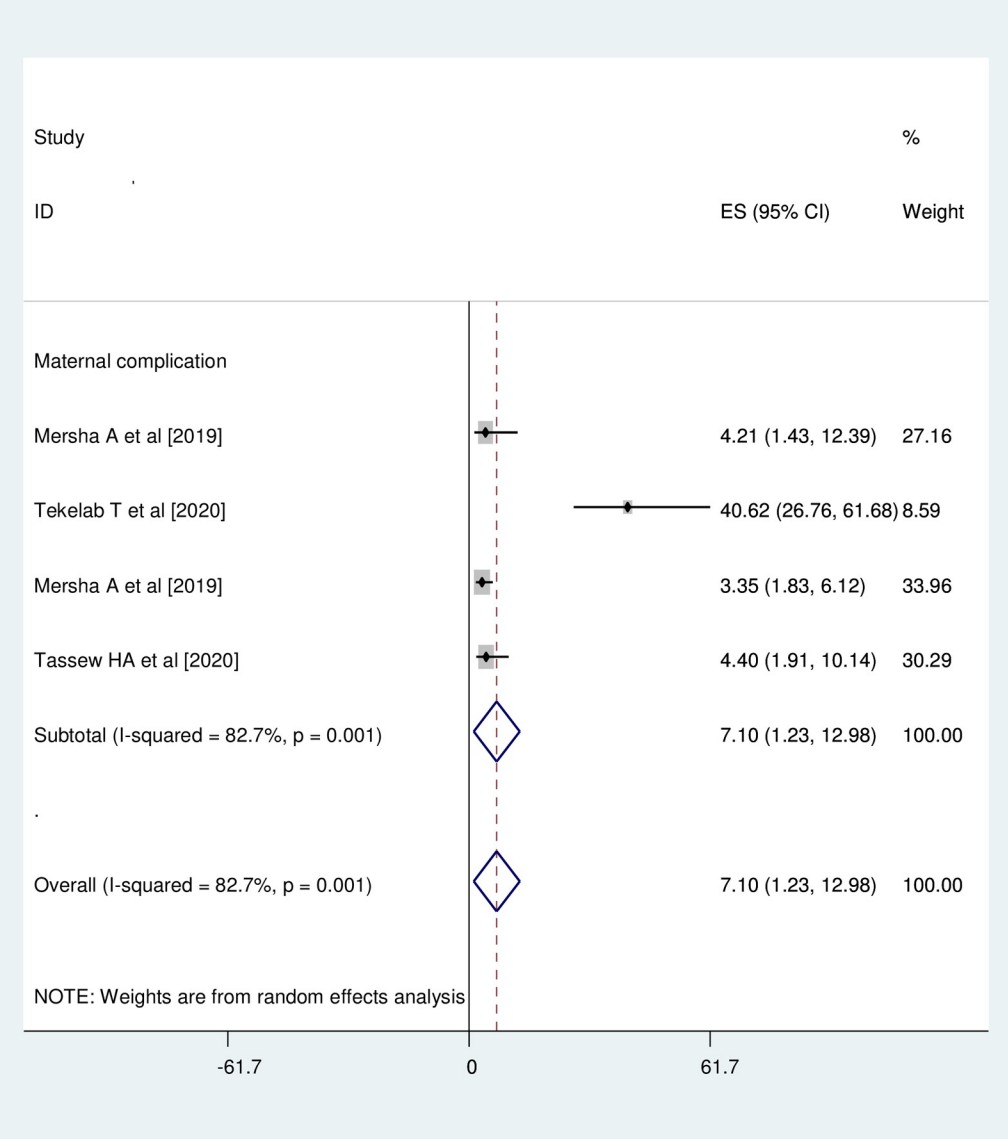

**Fig 8. Association between history of maternal medical complication and neonatal near miss.**

Obstructed labor may be risk factor for prolonged labor which in turn can lead to meconium staining. Various studies [25–30] reported that meconium staining is determinants of perinatal asphyxia. Neonates with perinatal asphyxia might fulfill the criteria for definition of neonatal near miss [1] since their management may involve parenteral antibiotic therapy, nasal CPAP and intubation. These all may be the reason behind for the association between obstructed labor and neonatal near miss. This association implies as neonates born after obstructed labor should get close supervision and better care.

Maternal medical complication that was found to be determinant factor of neonatal near miss in this study is supported with finding from other studies [7, 9, 12]. So, health care professional shall pay better attention of for newborns of these mothers in unit of neonatal care. Evidences [31, 32] reported that complications like hypertension during pregnancy leads to

preterm birth which is one of the criteria to define neonatal near miss [1] and this can be taken in to account for the current association.

## Limitations

Neonatal near miss had no clear standard definition and this challenged the inclusion of the articles. In addition, it is not widely studied in the country and this resulted in finding of few published articles. On the other hand, variables were not uniformly studied across the studies and it was difficult to match the variables from different studies for the Meta-Analysis. Besides; though we have tried to minimize it, the risk of bias may have overestimated the prevalence of near miss. Moreover, we couldn't review unpublished articles from universities in the country and this may result in very few missing.

## Conclusion

From the Meta-Analysis, the prevalence of neonatal near miss in Ethiopia is evidenced to be high. Primiparity, referral linkage, premature rupture of membrane, obstructed labor and maternal medical complications during pregnancy were found to be determinant factors of neonatal near miss in this study.

## Supporting information

**S1 File.**
(DOCX)

## Acknowledgments

We are grateful to Hawassa University for providing internet services which is vital for accomplishment of this systematic Review and Meta-Analysis. Again, we extend our thanks to Dr. Ayalew Astatkie for his support in reviewing. Moreover, our gratitude extends to Dr. Dereje Bayissa for his support in English edition.

## Author Contributions

**Conceptualization:** Ababe Tamirat Deressa, Melese Siyoum Desta.

**Data curation:** Ababe Tamirat Deressa, Melese Siyoum Desta.

**Formal analysis:** Ababe Tamirat Deressa, Melese Siyoum Desta.

**Methodology:** Ababe Tamirat Deressa, Melese Siyoum Desta.

**Software:** Ababe Tamirat Deressa, Melese Siyoum Desta.

**Validation:** Ababe Tamirat Deressa, Melese Siyoum Desta.

**Writing – original draft:** Ababe Tamirat Deressa.

**Writing – review & editing:** Ababe Tamirat Deressa, Melese Siyoum Desta.

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
