## [Decision Letter · Decision Letter 0]

15 Dec 2021

PONE-D-20-32942Prevalence and determinants of neonatal near miss in Ethiopia: A Systematic Review and Meta-AnalysisPLOS ONE

Dear Dr. Deressa,

Thank you for submitting your manuscript to PLOS ONE. After careful consideration, we feel that it has merit but does not fully meet PLOS ONE’s publication criteria as it currently stands. Therefore, we invite you to submit a revised version of the manuscript that addresses the points raised during the review process.

We look forward to receiving your revised manuscript.

Kind regards,

Carla Betina Andreucci Polido, M.D., P.h.D.

Academic Editor

PLOS ONE

Journal Requirements:

2.  Please attach a Supplemental file of the results of the individual components of the quality assessment, not just the overall score, for each study included. Please also explain the reasons, and number of studies excluded for each reason, in the flow diagram. Thank you.

3.  At this time, we ask that you please provide the full search strategy and search terms for at least one database used as Supplementary Information.

4.  Please include the date ranges for your search in the methods section.

6. Please upload a new copy of Figure 3 as the detail is not clear. Please follow the link for more information: https://blogs.plos.org/plos/2019/06/looking-good-tips-for-creating-your-plos-figures-graphics/" https://blogs.plos.org/plos/2019/06/looking-good-tips-for-creating-your-plos-figures-graphics/

8. We note that this manuscript is a systematic review or meta-analysis; our author guidelines therefore require that you use PRISMA guidance to help improve reporting quality of this type of study. Please upload copies of the completed PRISMA checklist as Supporting Information with a file name “PRISMA checklist”.

Reviewers' comments:

Reviewer's Responses to Questions

**Comments to the Author**

1. Is the manuscript technically sound, and do the data support the conclusions?

Reviewer #1: Yes

Reviewer #2: Yes

Reviewer #3: Partly

Reviewer #4: Partly

2. Has the statistical analysis been performed appropriately and rigorously? 

Reviewer #1: Yes

Reviewer #2: Yes

Reviewer #3: Yes

Reviewer #4: I Don't Know

3. Have the authors made all data underlying the findings in their manuscript fully available?

Reviewer #1: Yes

Reviewer #2: Yes

Reviewer #3: No

Reviewer #4: Yes

4. Is the manuscript presented in an intelligible fashion and written in standard English?

Reviewer #1: No

Reviewer #2: Yes

Reviewer #3: No

Reviewer #4: Yes

5. Review Comments to the Author

Reviewer #1: This is a Systematic Review and Meta-Analysis on prevalence and determinants of neonatal near miss in Ethiopia. The subject is current and relevant, and overall, the methods are adequate. However, the manuscript could have more robustness.

See please some comments below.

There is need for an English revision.

Abstract: adequate.

Introduction: I suggest you explain near miss criteria in the methods section, and not in your first paragraph. I suggest you explain a little further the importance of considering neonatal near miss together with neonatal deaths to expand health surveillance. I suggest you develop the text a little further, making hypotheses about the meaning of different available data on the prevalence of neonatal near miss worldwide, rather than simply presenting numbers. I also suggest that you describe the situation regarding infant deaths in Ethiopia, therefore highlighting the relevance of your research.

Methods: Your exclusion criteria are not clear. Was English the only language included in your search? I think the table with included studies is misplaced within the manuscript, it should appear after the PRISMA chart. I would like you to detail further how you handled the risk of bias in meta-analysis with case-control and cohort studies. You have mentioned a third reviewer handling conflicts of article inclusion in your systematic review, but the manuscript has two authors only. I suggest you name the third reviewer in acknowledgement section.

Discussion: Please explain why better socio-economic status contribute for better obstetric and neonatal care (funding for health programmes and facilities? Availability of childbirth care providers? Etc). What do you think will consist in “adequate birth preparedness” for primipara women? I don´t agree with the assumption of meconium-stained liquor to be a reason for neonatal near miss after obstructed labour. Meconium may be the consequence of a protracted labour, so reasons for the prolonged second stage of labour should play a more important role in the matter. Please provide further examples of maternal complications found in included studies that lead to neonatal near miss (you have mentioned hypertension only).

The text in limitations section is unclear.

Reviewer #2: The discussion brought important reflections, which are in line with the objective.

The methodology was adequate, as well as the statistics were satisfactorily performed: statistics were rigorous for the type of analysis in question, especially because it is an extremely relevant subject and with few articles published in the area.

The language is intelligible. The references used are recent.

Reviewer #3: The manuscript needs extensive English revision.

Some parts are confusing, e.g, Figure 4 is the same as Figure 2. In Figure 2, 5 studies were included in the final analysys but only 3 studies were included in the meta analysis of near miss prevalence.

What are the reasons for the 69 exclusions? It is recommended by rhe PRISMA guideline that authors report the reasons for exclusion.

Where is Figure 1 with the funnel plot?

Discussion should be more elaborated as some findings are poorly explained such as the findings about primiparity.

The study has an evident risk of publication bias, and this is difficult to access with less than 10 studies. This risk of bias may have overestimated the prevalence of near miss and this must be discussed.

Reviewer #4: Relevant study. Appropriate title. In introduction, it is important to add prevalence studies in developed countries, outside the African continent, only Brazil was mentioned. In the method, record the period defined for searching the databases. I recommend keeping table 1 of the characterization of the studies in the results section. One of the 5 studies selected to make up the review was not published, so how did researchers gain access to the full text?

The authors show in the method that they applied statistic tests to verify the heterogeneity but they do not present the test results.

6. PLOS authors have the option to publish the peer review history of their article (what does this mean?). If published, this will include your full peer review and any attached files.

Reviewer #1: No

Reviewer #2: No

Reviewer #3: No

Reviewer #4: No

---

## [Author Response · Author response to Decision Letter 0]

22 Jun 2022

Date: June, 2022

 To: PLOS ONE

Author’s Point-by-point responses

I am so grateful for the constructive and teaching comments raised from the reviewers and I have given responses for comments and questions respectively. Besides, I have highlighted the correction in the main manuscript document with truck changes. 

Reviewer’s comments Author’s responses

Reviewer 1

There is need for an English revision.

Introduction: I suggest you explain near miss criteria in the methods section, and not in your first paragraph. I suggest you explain a little further the importance of considering neonatal near miss together with neonatal deaths to expand health surveillance. I suggest you develop the text a little further, making hypotheses about the meaning of different available data on the prevalence of neonatal near miss worldwide, rather than simply presenting numbers. I also suggest that you describe the situation regarding infant deaths in Ethiopia, therefore highlighting the relevance of your research. I would like to express my gratitude for your constructive comments. My responses are elicited below. 

The English revision for the manuscript was undertaken with Open Grammarly software and one fluent speaker of the language whose name is stated under the Acknowledgement. 

I thought the background information including the scientific definition of the subject matter shall be in the introduction part as introductory phase. This write-up is recommended by many researchers. Now, I have also incorporated it in the Methods section. 

The findings of the literatures were also hypothesized in the last paragraph of the introduction. 

The importance of considering neonatal near miss together with neonatal deaths to expand health surveillance was also described. 

Neonatal and infant deaths in Ethiopia were also highlighted. 

Methods: Your exclusion criteria are not clear. Was English the only language included in your search? I think the table with included studies is misplaced within the manuscript, it should appear after the PRISMA chart. I would like you to detail further how you handled the risk of bias in meta-analysis with case-control and cohort studies. 

You have mentioned a third reviewer handling conflicts of article inclusion in your systematic review, but the manuscript has two authors only. I suggest you name the third reviewer in acknowledgement section.

 � As it was stated in the Methods Section, after searching the articles, articles without full text or abstract, duplicated studies and anonymous reports were excluded. Also, articles or reports written in language different from English was also excluded on searching phase. 

Table of included studies is part of Methods section under sub-section of “Inclusion and exclusion Criteria”. But, PRISMA chart is part of the “Result” since it describes the included and excluded researches for the two objectives separately. 

One approach to avoid bias in Meta-Analysis, especially selection bias, is conducting rigorous systematic reviews. Thus tough reviews were undertaken. Again the quality assessment was performed for each study by paying attention for related issues. 

The third reviewer is already mentioned in the acknowledgment section.

Discussion: Please explain why better socio-economic status contribute for better obstetric and neonatal care (funding for health programmes and facilities? What do you think will consist in “adequate birth preparedness” for primipara women? I don´t agree with the assumption of meconium-stained liquor to be a reason for neonatal near miss after obstructed labour. Meconium may be the consequence of a protracted labour, so reasons for the prolonged second stage of labour should play a more important role in the matter. Please provide further examples of maternal complications found in included studies that lead to neonatal near miss (you have mentioned hypertension only).

The text in limitations section is unclear. � The discussion point was to justify that the country with better socio-economic status can have improved level of medical equipment that can directly contribute to provide better or quality obstetric and neonatal care. 

Adequate birth preparedness here is to intensify any readiness the pregnant mother can have to get timely use of skilled maternal and neonatal care, especially during childbirth

In the definition of neonatal near miss, one pragmatic criteria is birth Asphyxia. Thus, the discussion here is to show that when there is Meconium staining or aspiration more likely there will be birth Asphyxia which fulfills the diagnosis of neonatal near miss. On the other hand, the perspective from which you rationalized is also correct. 

Reviewer #2 � Thank you for your constructive review!

Reviewer #3

The manuscript needs extensive English revision.

Some parts are confusing, e.g, Figure 4 is the same as Figure 2. In Figure 2, 5 studies were included in the final analysys but only 3 studies were included in the meta analysis of near miss prevalence.

What are the reasons for the 69 exclusions? It is recommended by rhe PRISMA guideline that authors report the reasons for exclusion.

Where is Figure 1 with the funnel plot?

Discussion should be more elaborated as some findings are poorly explained such as the findings about primiparity.

The study has an evident risk of publication bias, and this is difficult to access with less than 10 studies. This risk of bias may have overestimated the prevalence of near miss and this must be discussed.

 I would like to express my gratitude for your constructive comments. My responses are elicited below. 

The English revision for the manuscript was undertaken with Open Grammarly software and one fluent speaker of the language whose name is stated under the Acknowledgement. 

Figure 2, corrected to Figure 1, presents the studies that are selected to determine the prevalence of neonatal near miss. In this Figure, the case control and cohort studies are not included. On the other side, Figure 4, corrected to Figure 3, presents the studies that are selected to determine the predictors of neonatal near miss. In this figure studies (including case-control and cohort study) that conducted the analysis for associated factors, determinants or predictors of neonatal near miss were included. 

Thank you for your correcting comment! Now I have corrected that 3 studies were included in Figure 2 to determine the pooled prevalence on neonatal near miss. 

As it was stated in the Methods Section, after searching the articles, articles without full text or abstract, duplicated studies and anonymous reports were excluded. Also, articles or reports written in language different from English was also excluded on searching phase. 

Again, thank you for your exceptional look. I haven’t excluded Figure 1 for invisibility of its graphics and I have now removed it from the manuscript. 

Other discussions are well looked. 

I agree with your idea concerning publication bias. Here it is analyzed to look its presence and absence. But, it is presence is not pronounced as limitation since the selected studies are less than ten. 

The possible effect of risk of bias on the result is elaborated in the Limitation section in this revised Manuscript. 

Reviewer #4 

Relevant study. Appropriate title. In introduction, it is important to add prevalence studies in developed countries, outside the African continent, only Brazil was mentioned. 

I recommend keeping table 1 of the characterization of the studies in the results section. One of the 5 studies selected to make up the review was not published, so how did researchers gain access to the full text?

The authors show in the method that they applied statistic tests to verify the heterogeneity but they do not present the test results.

 � Thank you very much for your constructive Review!

The comment is well incorporated and the literatures from developing countries are emphasized just to make sound and equivalent discussions with the current study.

Since it summarizes the number of selected studies from the search, hat is the reason for presenting Table 1 in Methods section. 

As it recommended one approach of minimizing selection bias is searching and including unpublished articles. Thus; we, the authors, included the unpublished articles from the library of Hawassa University where we were working. 

The Figure of heterogeneity test is removed due to its graphics invisibility. However, the test result is written in text in the paragraph.

---

## [Decision Letter · Decision Letter 1]

24 Aug 2022

PONE-D-20-32942R1Prevalence and determinants of neonatal near miss in Ethiopia: A Systematic Review and Meta-AnalysisPLOS ONE

Dear Dr. Deressa,

Thank you for submitting your manuscript to PLOS ONE. After careful consideration, we feel that it has merit but does not fully meet PLOS ONE’s publication criteria as it currently stands. Therefore, we invite you to submit a revised version of the manuscript that addresses the points raised during the review process.

The reviewers suggested further revision in your manuscript. The review comments can be found at the end of this email. Please note some minor clarifications needed and the suggested language revision.

Thank you for submitting your article to Plos One. We look forward to receiving your revision.

We look forward to receiving your revised manuscript.

Kind regards,

Carla Betina Andreucci, M.D., P.h.D.

Academic Editor

PLOS ONE

Journal Requirements:

Reviewers' comments:

Reviewer's Responses to Questions

**Comments to the Author**

1. If the authors have adequately addressed your comments raised in a previous round of review and you feel that this manuscript is now acceptable for publication, you may indicate that here to bypass the “Comments to the Author” section, enter your conflict of interest statement in the “Confidential to Editor” section, and submit your "Accept" recommendation.

Reviewer #2: All comments have been addressed

Reviewer #3: (No Response)

2. Is the manuscript technically sound, and do the data support the conclusions?

Reviewer #2: Yes

Reviewer #3: Yes

3. Has the statistical analysis been performed appropriately and rigorously? 

Reviewer #2: Yes

Reviewer #3: Yes

4. Have the authors made all data underlying the findings in their manuscript fully available?

Reviewer #2: Yes

Reviewer #3: Yes

5. Is the manuscript presented in an intelligible fashion and written in standard English?

Reviewer #2: Yes

Reviewer #3: No

6. Review Comments to the Author

Reviewer #2: The researchers accepted most of the suggestions, with a great improvement in the paper.

I believe that some topics of the "discussion" could have been discussed more comprehensively.

I reinforce the importance of the topic.

Reviewer #3: The manuscript has suffered many corrections, but English needs to be revised. Maybe there are texting problems (for example, from line 69-71 and from 126-127).

On line 89, is the period correct? from 15-25 august 2020 only?

On results, line 140-143, the sum does not match.

All figures need to be checked, for example, is figure 5 actually figure 4? is figure 6 actually figure 5?

On line 238-244, it must be emphasized that the bias of publication was a matter of concern.

7. PLOS authors have the option to publish the peer review history of their article (what does this mean?). If published, this will include your full peer review and any attached files.

Reviewer #2: No

Reviewer #3: No

---

## [Author Response · Author response to Decision Letter 1]

19 Oct 2022

Reviewer #2

The researchers accepted most of the suggestions, with a great improvement in the paper.

I believe that some topics of the "discussion" could have been discussed more comprehensively.

I reinforce the importance of the topic.

Author’s responses

I would like to express my gratitude for your constructive comments. Now, I have tried to address the concern and rearranged the paragraphs under discussion. 

Reviewer #3

The manuscript has suffered many corrections, but English needs to be revised. Maybe there are texting problems (for example, from line 69-71 and from 126-127).

On line 89, is the period correct? from 15-25 august 2020 only?

On results, line 140-143, the sum does not match.

All figures need to be checked, for example, is figure 5 actually figure 4? is figure 6 actually figure 5?

On line 238-244, it must be emphasized that the bias of publication was a matter of concern.

Author’s responses

Thank you for your deep looking and constructive comments. Now, the comments are well incorporated in the edited manuscript accordingly on their respective lines. Besides, number of articles under result section was edited and written in clear expression as follow. 

“A total of 101 articles and one article were found with electronic and other approach of searching respectively. Twenty eight were duplication and 74 articles were screened. From these 74 articles, 69 of them were excluded due to irrelevance, absence of full text or Abstract”.

August 15-25, 2020 was the period for on line data searching. Thus, it was correct.

---

## [Editor Report · Decision Letter 2]

23 Nov 2022

Prevalence and determinants of neonatal near miss in Ethiopia: A Systematic Review and Meta-Analysis

PONE-D-20-32942R2

Dear Dr. Deressa,

We’re pleased to inform you that your manuscript has been judged scientifically suitable for publication and will be formally accepted for publication once it meets all outstanding technical requirements.

Kind regards,

Carla Betina Andreucci, M.D., P.h.D.

Academic Editor

PLOS ONE

---

## [Editor Report · Acceptance letter]

29 Nov 2022

PONE-D-20-32942R2 

Prevalence and Determinants of Neonatal Near Miss in Ethiopia: A Systematic Review and Meta-Analysis 

Dear Dr. Deressa:

I'm pleased to inform you that your manuscript has been deemed suitable for publication in PLOS ONE. Congratulations! Your manuscript is now with our production department. 

Kind regards, 

on behalf of

Mrs. Carla Betina Andreucci 

Academic Editor

PLOS ONE